# Controlled ultrafast ππ*-πσ* dynamics in tryptophan-based peptides with tailored micro-environment

Marius Hervé [1], Alexie Boyer [1], Richard Brédy [1], Isabelle Compagnon[1], Abdul-Rahman Allouche [1] & Franck Lépine [1✉]

Ultrafast charge, energy and structural dynamics in molecules are driven by the topology of the multidimensional potential energy surfaces that determines the coordinated electronic and nuclear motion. These processes are also strongly influenced by the interaction with the molecular environment, making very challenging a general understanding of these dynamics on a microscopic level. Here we use electrospray and mass spectrometry technologies to produce isolated molecular ions with a controlled micro-environment. We measure ultrafast photo-induced ππ*-πσ* dynamics in tryptophan species in the presence of a single, charged adduct. A striking increase of the timescale by more than one order of magnitude is observed when changing the added adduct atom. A model is proposed to rationalize the results, based on the localized and delocalized effects of the adduct on the electronic structure of the molecule. These results offer perspectives to control ultrafast molecular processes by designing the micro-environment on the Angström length scale.

[1] Univ Lyon, Université Claude Bernard Lyon 1, CNRS, Institut Lumière Matière, Villeurbanne, France. ✉email: franck.lepine@univ-lyon1.fr

Ultrafast processes triggered by light absorption in biomolecules have prime importance in nature[1–3]. The complex interplay between electronic and nuclear degrees of freedom is responsible for information and energy transport, photoprotection, and structural changes, therefore essential to life sustainability. Understanding these processes at the microscopic atomic scale requires ultrafast pump–probe technology in order to determine the timescales as well as the degrees of freedom involved, and overall to decipher the multiple-step mechanisms induced by light[4]. To understand how nature has built these elementary processes, gas phase experiments represent a powerful tool as it allows to study the intrinsic properties of the molecule with a controlled (or absent) environment. Furthermore, experimental results can serve as a benchmark for high-level quantum chemistry calculations, eventually leading to a complete multiscale description of the link between electronic, geometrical structures, and reaction path.

Current ultrafast pump–probe experiments on gas-phase complex molecules are mostly performed on neutral compounds, demonstrating the prominence of this technique to unravel the subtle mechanisms at play in, e.g., ultrafast charge dynamics[5], structural rearrangement[6,7], and energy dissipation[8–10], all involving non-adiabatic electro-nuclear dynamics. For biomolecules however, species are most often found in ionic forms (deprotonated, protonated, or complexed with a charged adduct) in biological conditions, where the charge state has a major influence on the chemical landscape and molecular reactivity. The development of ElectroSpray Ionization (ESI) technology has enabled the production of such charged molecules into the gas phase, opening new perspectives in the investigation of ionic species and in the development of relevant ultrafast experimental schemes[11].

Ultrafast molecular processes are often driven by electronic or nuclear degrees of freedom as well as their couplings. In the Born–Oppenheimer representation, electronic states are defined along nuclear coordinates, and non-adiabatic dynamics occur at energy degeneracies where states cross[12]. A seminal example concerns the coupling between two states of different character, as the so-called $\pi\pi^*$–$\pi\sigma^*$ states, where the $\pi\pi^*$ state corresponds to the excitation of typical aromatic chromophores, and the $\pi\sigma^*$ state has a $\sigma$ character located on a specific X-H bond, with X being usually a heteroatom (N, O, or S). This prototypical situation is responsible for charge and energy transfer in aromatic based biologically relevant molecules[13]. For instance, experiments in aromatic species[14,15], peptides containing aromatic amino acids[16,17] or nucleotides[18] have shown that photo-induced mechanisms driven by $\pi\pi^*$–$\pi\sigma^*$ couplings usually occur on very fast timescales.

Beyond the study of the natural processes occurring on ultrafast timescales, new possibilities to manipulate photo-induced reactions arise, offering new perspectives in chemistry or material science. Control strategies are numerous, spanning from coherent control that uses specially designed laser pulses to manipulate population transfer and couplings[19], to specific bond excitation using infrared (IR) light in the particular case of charge transfer[20], or chemical design where the atomic composition of the molecule is changed in order to improve a targeted photophysical property[21].

Here we demonstrate that tailoring the micro-environment at the atomic scale in isolated amino acids and peptides allows us to tune the timescale of the $\pi\pi^*$–$\pi\sigma^*$ dynamics by more than one order of magnitude. We used a pump–probe scheme to measure this timescale variation in tryptophan-containing species as a function of a controlled micro-environment, comprising an adduct atom. Combined with quantum chemistry calculations, we show that this effect is due to the tuning of the relative energy between the $\pi\pi^*$ and $\pi\sigma^*$ states, that is induced by the interaction with the adduct. This mechanism is deciphered by considering the separate effect of the adduct atom on the two states. It allows us to rationalize the behavior of other biomolecular structures as demonstrated in the case of a dipeptide, a first step towards more complex environments.

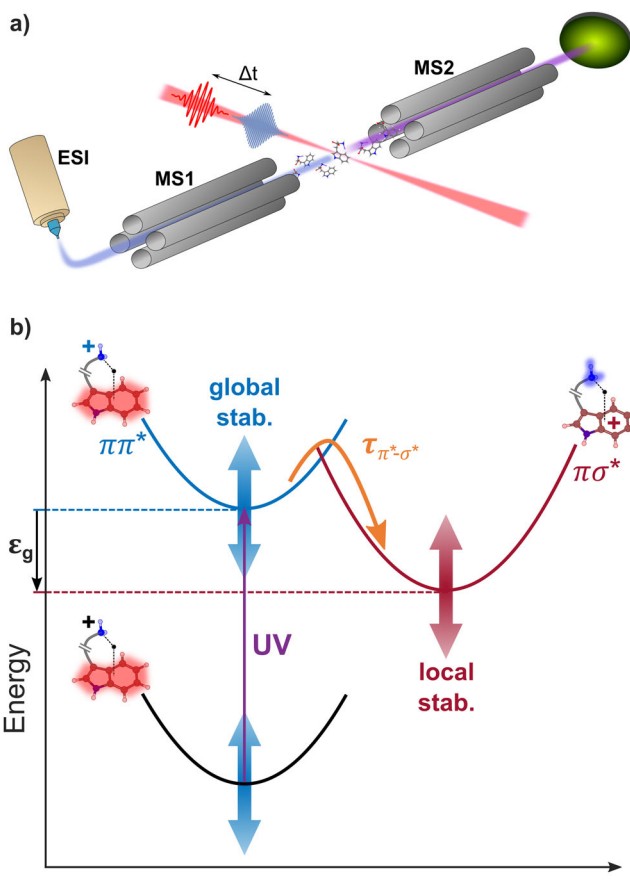

**Fig. 1 Global and local stabilization effects in $\pi\pi^*$-$\pi\sigma^*$ dynamics of molecular ions. a** Experimental setup for time-resolved spectroscopy of on-the-fly molecular ions, combining an electrospray source (ESI), mass selection (MS1), laser interaction, and detection (MS2). **b** Concept of Global-Local Stabilization Separation (GLoSS), illustrated in the case of tryptophan-containing species: after absorption (in the UV range in this work), the relative energy gap $\varepsilon_g$ between the excited $\pi\pi^*$ state (in blue, the red shading indicates the excited electron density) and the $\pi\sigma^*$ state (in red, the blue shading indicates the excited electron density) can be tuned to directly change the $\pi\pi^*$-$\pi\sigma^*$ transfer timescale. More precisely, the two states can be changed in energy by playing separately on the complexation interaction of the molecule with the adduct atom (global stabilization acting on the ground state) or on the specific interaction of the adduct with the $NH_2$ moiety (local stabilization), thus changing their energy gap (in this figure, the $\pi\sigma^*$ state is represented as a bond state for sake of simplicity, while it corresponds to a dissociative state).

## Results and discussion

**Experiment**. A schematic representation of the experimental setup is presented in Fig. 1a. It consists of a triple-quadrupole mass spectrometer instrument coupled to ultrafast laser technology[22]. In brief, molecular ions are produced in the gas phase using an ESI source. The ions are then filtered according to their mass-to-charge ratio by a first quadrupole (MS1). This

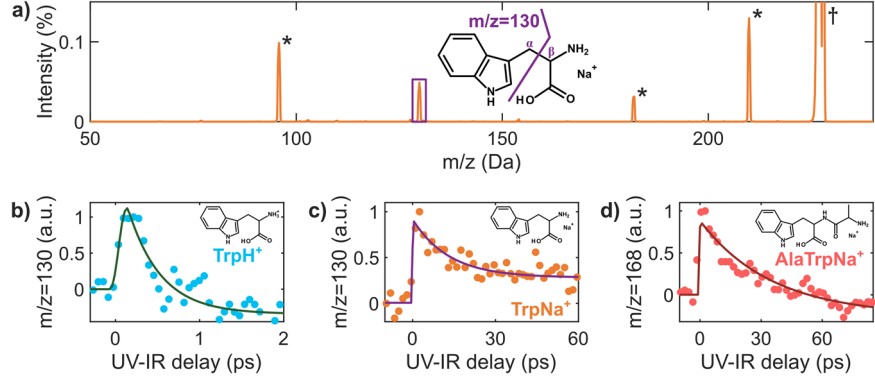

**Fig. 2 Time-resolved ππ\*–πσ\* dynamics in tryptophan-containing charged peptides. a** Mass spectrum of sodiated tryptophan ($m/z = 227$, indicated by the † marker), obtained after UV irradiation. The * markers depict residual (CID) fragments, and the structure of fragment $m/z = 130$ ($C_\alpha$-$C_\beta$ cleavage in purple) is shown. Intensity on the y-axis represents the signal normalized on the parent intensity (i.e., $m/z = 227$ is 100%). **b** Normalized time-dependent yield of fragment $m/z = 130$ in TrpH$^+$, as a function of the UV pump–IR probe delay (blue dots), together with its exponential fit (dark green line), yielding a timescale of $390 \pm 100$ fs. **c, d** Same analysis for TrpNa$^+$ (orange dots) and AlaTrpNa$^+$ (red dots), giving timescales of $13 \pm 3$ ps (purple line) and $35 \pm 8$ ps (dark red line) respectively. The chemical structure of the three molecules is shown in each box. In the case of AlaTrpNa$^+$, fragment $m/z = 130$ is not observed, and the dynamics is observed in, e.g., fragment $m/z = 168$, as shown in (**d**) and in Supplementary Fig. 3.

allows the selection of the molecular ions of interest, which then interact with the laser pulses. The products of the interaction are analyzed by a second quadrupole mass filter (MS2). We used the output of a 25 fs, 800 nm, 2 mJ, 5 kHz laser to produce a pump ultraviolet (UV) pulse at 267 nm by third-harmonic generation. It is recombined with the 800 nm IR probe beam for time-resolved experiments. A typical pump–probe measurement consists of recording mass spectra obtained with MS2 at each UV–IR pump–probe delay. The dynamics is monitored by measuring the delay-dependent variation of the fragment signal. To do so the signal obtained following the interaction with the UV pump only is subtracted from the signal obtained with the pump and the probe, for each delay. Compared to other existing configurations using an ion trap, the perpendicular, "on-the-fly", design ensures that molecular ions interact only once with the pump and probe pulses, therefore simplifying the interpretation.

The mass spectrum obtained following photoexcitation of sodiated tryptophan (TrpNa$^+$, $m/z = 227$) at 267 nm is presented in Fig. 2a. It shows that laser interaction with the ions dominantly leads to photofragment $m/z = 130$, which is barely present in statistical dissociation experiments such as collision-induced dissociation (CID)[23,24]. This UV-induced fragment corresponds to $C_\alpha$–$C_\beta$ cleavage as indicated in Fig. 2a[25], and shows that specific dynamics are at play after absorption to the first excited states.

In order to measure the dynamics induced by UV absorption, we performed UV–IR pump–probe measurements with our on-the-fly apparatus. We recorded the variation of the $m/z = 130$ photofragment yield as a function of the pump–probe delay, as shown in Fig. 2c. A clear time-dependent signal is obtained showing a very fast rise of the signal around the UV–IR temporal overlap followed by an exponential decay in the picosecond time domain, which eventually leads to a steady contribution (step). In the experiment, the pump pulse excites the parent ππ\* state, and the probe pulse promotes the population of the excited state to higher lying states, eventually leading to fragmentation. The delay-dependent signal of the $m/z$ 130 photofragment is thus a signature of the dynamics in the ππ\* state. Let us notice that this fragment is simply a convenient signature of the dynamics occurring in the parent ππ\* excited state, as such that the extracted timescale is related to the excited state dynamics and not to the fragmentation process itself.

Fitting the data using a single exponential model yielded $\tau = 13 \pm 3$ ps. Following the same procedure, we studied the dynamics

in two other tryptophan-containing systems: protonated tryptophan (TrpH$^+$) and sodiated alanyl-tryptophan (AlaTrpNa$^+$). These two molecules are built on the same chromophore as TrpNa$^+$ (the indole ring) but with modifications in the ion adduct or the peptide chain. The results of the time-dependent measurements are presented in Fig. 2b (TrpH$^+$) and Fig. 2d (AlaTrpNa$^+$). In the case of TrpH$^+$, the same fitting procedure yielded a timescale of $390 \pm 100$ fs, in very good agreement with results reported in the literature ($380 \pm 50$ fs)[26–28]. For AlaTrpNa$^+$, the extracted timescale is $35 \pm 8$ ps. In all cases, a similar behavior is obtained, corresponding to a fast dynamics driven by a single decay (see Section 1 of Supplementary Methods and Supplementary Figs. 1–3 for more details). However, the extracted timescale becomes slower and slower when changing from TrpH$^+$ to TrpNa$^+$ (the decay time increases by a factor 30) and to AlaTrpNa$^+$ (increase by a factor 100), the timescale being thus tuned on two orders of magnitude. We note that this striking variation is obtained by simply changing the local environment of the indole chromophore, without affecting the molecular backbone.

## Calculations

In order to rationalize the observed trend, excited-state calculations were performed on the three molecular compounds. Indeed, UV absorption of indole-based molecules is governed by ππ\* transitions, where the excited electron density is located on the indole ring. Following geometry optimization using density functional theory (DFT) at the B3LYP/6-311+G(d,p) level of theory, RICC2/aug-cc-pVDZ calculations were conducted for the energy of the electronic excited states, as well as calculations of their related potential energy surfaces (PESs). The result for the excited states is presented in Fig. 3, together with the ground state geometries. ππ\* states are identified as the blue states, whose oscillator strength is the strongest. Additionally, two different states are present, with a πσ\* and ππ$_{CO}$\* electronic character, respectively. In the three molecules, the πσ\* state is localized on the NH$_2$X moiety, and lies in the vicinity of the excited ππ\* states. One can thus expect ultrafast dynamics to occur due to non-adiabatic couplings between these two states. To confirm the presence of such couplings, we computed the PESs for TrpNa$^+$. We studied the reaction along the N–X bonds, as they are known as a major pathway in ππ\*–πσ\* dynamics[14]. PESs reported in

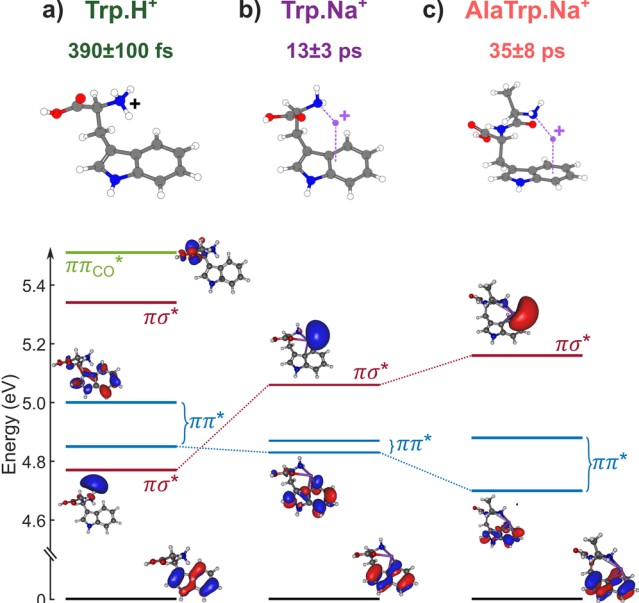

**Fig. 3 Excited electronic states in tryptophan-containing charged peptides.** Calculated vertical excited states for TrpH$^+$ (**a**), TrpNa$^+$ (**b**), and AlaTrpNa$^+$ (**c**), together with their equilibrium geometry (on top). The corresponding dominant orbitals are also displayed, giving the state-character of the excited state ($\pi\sigma^*$ states in red, $\pi\pi^*$ states in blue, $\pi\pi_{CO}^*$ state in green, with the ground state being in dark). Changing the adduct atom (from **a**) to **b**)) or changing the molecular backbone (from b) to c)) has a different influence on the position of the $\pi\sigma^*$ states and $\pi\pi^*$ states.

Fig. 4 show that a non-adiabatic crossing occurs between the $\pi\pi^*$ and $\pi\sigma^*$ states: while the $\pi\sigma^*$ state is bound along the N–H coordinate (Fig. 4a), it is dissociative along the N-Na coordinate, crossing the $\pi\pi^*$ state (Fig. 4b). Moreover, the dissociative character of the $\pi\sigma^*$ state is responsible for the specific fragmentation pattern observed in the experiment, on longer timescales.

The described mechanism is very similar for the two other species (see section 4 of Supplementary Methods and Supplementary Figs. 6, 7 for other PESs). Thus, as shown schematically in Fig. 1b, the observed dynamics corresponds to the $\pi\pi^*$–$\pi\sigma^*$ non-adiabatic coupling for the three molecules, enabling a population transfer to the $\pi\sigma^*$ state following local excitation of the indole ring. Within this framework, the transfer timescale $\tau$ is expected to strongly depend on the energy gap $\varepsilon_g$ between the two states according to simple energy gap considerations. This is indeed observed in our calculations (Fig. 3): when changing from TrpH$^+$ to AlaTrpNa$^+$, little change occurs in the position of the $\pi\pi^*$ states while a drastic increase in the energy of the $\pi\sigma^*$ state is noticed (from 4.77–5.16 eV). Consequently, the striking increase of the timescale is a direct consequence of the increase in the energy gap between the coupled states, due to changes in the micro-environment, as observed for substituted neutral heteroatomic molecules[29–32] or water-ion complexes[33]. Let us note that the reported energies are vertical energies from the ground state minimum energy geometry. Although the adiabatic energies usually serve as a reference when considering non-adiabatic dynamics, the choice of vertical energies allows us to have the same reference for all molecules even in the case of dissociative states or more complex PES.

To go deeper in the understanding of this effect, it is instructive to understand the key parameters imposed by the environment. For a well-defined molecular system (tryptophan-based ions in this study), the effect of the adduct atom can be understood in

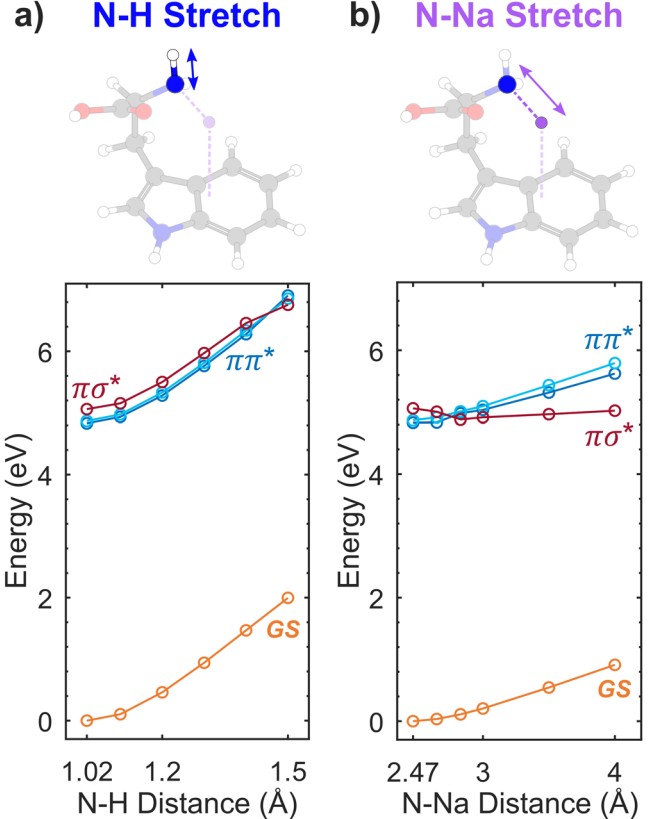

**Fig. 4 Reactive potential energy surfaces in sodiated tryptophan.** Computed potential energy surfaces for TrpNa$^+$, along the N–H (**a**) and N-Na (**b**) stretching modes, depicted by the geometries on top. The ground state equilibrium geometry corresponds to $R_{NH}$ = 1.02 Å and $R_{NNa}$ = 2.47 Å. The $\pi\sigma^*$ state is displayed in red, $\pi\pi^*$ states are displayed in blue, and the ground state is displayed in orange.

terms of energetic stabilization of the electronic states involved. Here, the excited $\pi\pi^*$ state corresponds to an electron density localized on the UV chromophore, i.e., the indole ring, as in the ground state. This means that both the ground and $\pi\pi^*$ states will be sensitive to the global interaction of the molecule with the adduct atom. In contrast, the $\pi\sigma^*$ state corresponds to an electron density localized on the N-terminus, i.e., its localization differs from the ground and $\pi\pi^*$ states. It is therefore sensitive to the local interaction with the adduct atom, because the electron density in this state becomes sensitive to the electrostatic interaction in the region of the NH$_2$X moiety.

In this picture, the increase of the energy gap observed in the results of the RICC2/aug-cc-pVDZ calculations can be rationalized by global and local stabilization of the $\pi\pi^*$ and $\pi\sigma^*$ states when changing the adduct atom. The energy change of the $\pi\pi^*$ state induced by the global effect of the adduct depends mainly on the global affinity of Trp for the considered adduct X$^+$, called XA thereafter. On the other hand, the energy change of the $\pi\sigma^*$ state due to local effects of the adduct is mainly determined by interactions between the NH$_2$ moiety and the adduct X$^+$ (Fig. 1b). In a first approximation, this interaction is considered as a long-range Coulomb attraction ($E_{Coul}$, see Supplementary Discussion for more details). These simple considerations lead to the following expression for the energy gap increase:

$$\delta\varepsilon_g = \Delta XA - \Delta E_{Coul} = \Delta XA - \frac{e^2}{4\pi\varepsilon_0}\Delta\left[\frac{1}{R_{NX}}\right] \qquad (1)$$

where $\varepsilon_g = E_{\pi\sigma^*} - E_{\pi\pi^*}$ is the (relative) energy gap between the $\pi\sigma^*$ state and the $\pi\pi^*$ state for a given species, associated to a specific chromophore (here the indole ring). $\delta\varepsilon_g$ is the variation of the energy gap when changing from one adduct to another, e.g., from H to Na. $\Delta XA$ and $\Delta E_{Coul}$ are respectively the changes in affinity for the adduct atom X and in Coulomb energy when changing the adduct. Finally, $R_{NX}$ is the equilibrium distance between the adduct X and the $NH_2$ moiety. For the specific case of $TrpH^+$ and $TrpNa^+$, considering the different affinities[34] and equilibrium distances for the proton and sodium adducts, this leads to an increase of the energy gap of $\delta\varepsilon_g = 395$ meV from $TrpH^+$ to $TrpNa^+$, which is strikingly close to the result of the ab initio calculations, that predict an increase of $\delta\varepsilon_g = 310$ meV.

Based on this simple model it becomes straightforward to understand the process in terms of the combined global and local effects of the adduct atom on the molecule. This Global-Local Stabilization Separation model (GLoSS) is very informative since it enables to disentangle each contribution and how it will influence the timescale of the dynamics: increasing the global adduct affinity will increase the timescale by increasing the energy gap, while a closer $NH_2$–X distance decreases the timescale by increasing the Coulomb stabilization of the $\pi\sigma^*$ state.

The GLoSS model can be further illustrated in the case of $AlaTrpNa^+$. In this case, the adduct atom remains unchanged ($Na^+$) compared to $TrpNa^+$, but the environment of the indole chromophore becomes more flexible because of the alanyl residue. Thus, the global complexation of the $Na^+$ adduct is more favorable than for $TrpNa^+$, resulting in an increase of the sodium affinity[35], and based on the GLoSS model it should increase further the $\pi\pi^*$–$\pi\sigma^*$ timescale, which is indeed observed (Fig. 2). This increase arises from the separated stabilization of the $\pi\pi^*$ state by complexation, while the $\pi\sigma^*$ state is unaffected because the $NH_2$-Na distance remains constant between $TrpNa^+$ and $AlaTrpNa^+$. Again, the GLoSS model gives a good agreement with ab initio calculations ($\delta\varepsilon_g = 326$ meV when $AlaTrpNa^+$ is compared to $TrpNa^+$, instead of 230 meV in the calculations), showing a clear illustration of the separation of the stabilization contributions.

While this $\pi\pi^*$–$\pi\sigma^*$ dynamics is associated with non-adiabatic couplings between the two states, it can also be seen as a charge transfer (CT) process. Because of the difference in the density localization of the states, the dynamics implies an electron transfer from one side of the molecule to another, as can be clearly seen in Fig. 3. Such a CT process was also invoked in the relaxation of protonated tryptophan[26,27]. In this scenario, the species absorb UV light through the $\pi\pi^*$ excitation localized on the indole chromophore and it has been shown that a CT process might occur through the $\pi\pi^*$-$\pi\sigma^*$ coupling described in this work. The initial positive charge, borne by the $NH_3^+$ terminus, transits to the indole moiety when the population is transferred to the $\pi\sigma^*$ state[36,37]. In the case of sodiated species studied here, the same process occurs, showing that CT dynamics can be controlled using the micro-environment.

In conclusion, we have shown that $\pi\pi^*$–$\pi\sigma^*$ dynamics can be controlled by using the influence of an adduct atom positioned on the Angström length scale. On a quantum mechanical point of view, the three molecules investigated have similar electronic structures, which ensures that the mechanism can be described by a limited number of involved states. The influence of the adduct is rationalized as a tuning of the relative energy gap between the $\pi\pi^*$ and the $\pi\sigma^*$ states that strongly influences the non-adiabatic coupling efficiency, and thus the femtosecond-to-picosecond timescale. This mechanism is general and can be understood as the mixed contribution of the role of global and local stabilization effects. These contributions can be disentangled in a GLoSS model based on first principle arguments. Such a model, where

the different stabilization contributions are separated, could be further used to molecular systems with states of different electronic characters, as performed here for $\pi\pi^*$–$\pi\sigma^*$ dynamics. In turn, it offers simple control strategies based on atomic length scale designed micro-environment of more or less flexible molecular structures to influence the charge and energy flow. Although this work focuses on the influence of adduct atoms on $\pi\pi^*$–$\pi\sigma^*$ dynamics in peptides, the UV-induced dynamics in these species is rather complex and many other aspects might be investigated in the future. More generally, other processes such as structural changes, including isomerization, or spin effects (role of triplet states) might also be considered in the perspective of a controlled micro-environment, calling for new experimental investigations.

## Methods section

**Experimental section.** Pump–probe measurements were conducted using a UV–IR interferometer. The laser output (800 nm, 25 fs, 2 mJ, 5 kHz) was first used to generate 267 nm photons by third-harmonic generation (using two BBO crystals of 200 μm and 50 μm thicknesses). The UV pump beam is then separated from the residual IR light by a dichroic beamsplitter. It passes through a reflective delay line, before being recombined with the IR probe beam using a second dichroic beamsplitter. In this experiment, the IR probe beam consists of the residual IR that was not converted into UV photons. Its energy is controlled using a half-wave plate and a polarizer, and a second half-wave plate is used to control the polarization. Both beams are focused collinearly using an $f = 1$ m lens on the ion beam, which is produced by the triple-quadrupole instrument. The IR probe focus was kept 10 cm away from the ion beam, to fully overlap the UV beam spot and avoid extensive ionization. The UV and IR energies were 5 μJ and 800 μJ, respectively, yielding an IR intensity of $1-2 \times 10^{12}$ W cm$^{-2}$. Due to this low intensity, no ionization nor fragmentation induced by the IR pulse alone was detected, and the probe mechanism thus consists of perturbative absorption to higher lying excited states of the molecules. For each pump–probe delay, an average mass spectrum was recorded over 40 s.

After integration of the yield of the different photofragments for each delay, a fitting procedure was used, using the following formula:

$$\Delta S(t) = \theta(t - t_0) \cdot \left[ A_{decay} \cdot exp\left(-\frac{t - t_0}{\tau}\right) + A_{step} \right] \quad (2)$$

where $\tau$ is the lifetime of the exponential decay, $t_0$ the delay at which the pump and probe pulses overlap temporally, $A_{step}$ the amplitude difference between the yield at infinite negative and positive delays, $A_{decay}$ the amplitude of the decay, and $\theta$ the Heaviside function. For $AlaTrpNa^+$, the same $(\tau, t_0)$ parameters were fitted for all the fragments in a multidimensional procedure. In the case of $TrpH^+$, the signal was fitted by the convolution of Eq. (2) and a gaussian cross-correlation, due to finite pulse durations. Error bars of the extracted lifetime correspond to standard deviation of the least-square fitting procedure.

Chemicals were purchased from Sigma-Aldrich, and solutions were prepared at a concentration of 200 μM in a 50:50 mixture of $MeOH:H_2O$. 0.1% of acetic acid was added to the solution for the production of $TrpH^+$, and NaCl was added to a concentration of 0.1 mM for the production of $TrpNa^+$ and $AlaTrpNa^+$.

**Computational methods.** Geometries of the ground states of $TrpH^+$, $TrpNa^+$, and $AlaTrpNa^+$ were optimized by DFT calculations using the B3LYP functional at the 6-311+G(d,p) level (see Supplementary Tables 6–8 for the atomic coordinates). For the most stable conformer of each species, the excited states have been calculated using RICC2 computations, at the aug-cc-pVDZ

level, known to give good results in these protonated systems[11,37,38]. As mentioned in Sections 2–3 of the Supplementary Methods, we also conducted these calculations for higher-energy conformers (Supplementary Figs. 4–5 and Supplementary Tables 1, 3) as well as for different basis sets and geometries (Supplementary Table 2 and Supplementary Fig. 8), where the overall energetic position of the states remains similar to the most stable conformer. Finally, the potential energy surfaces were constructed by calculating the excited states while stretching the considered bond, without further geometry optimization. Further details about the geometries, excited states and PESs can be found in Sections 2–4 of the Supplementary Methods. Details about the results and derivation of the GLoSS model are given in Supplementary Discussion, Supplementary Figs. 9–10 and Supplementary Tables 4, 5.

## Data availability
The data that support the findings of this study are available from the corresponding author upon reasonable request.

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

## Acknowledgements
The authors thank Gilles Grégoire and Satchin Soorkia for fruitful discussions concerning the experimental results, the interpretation and for providing NPA calculations. The authors thank Baptiste Schindler, Vincent Loriot, and Eric Constant for their help on the technical implementation. The research has been supported by CNRS, ANR-16-CE30-0012 "Circé". In this work, we were granted access to the HPC resources of the FLMSN, "Fédération Lyonnaise de Modélisation et Sciences Numériques", partner of EQUIPEX EQUIP@MESO and to the "Centre de calcul CC-IN2P3" at Villeurbanne, France.

## Author contributions
M.H., I.C. and F.L. conceived the project. M.H., A.B., R.B., I.C. and F.L. conducted the experiments. M.H. and A.B. performed the data analysis. A.A. performed the calculations and analyzed it together with M.H.A.A. and F.L. developed the interpretation model and wrote the manuscript, with inputs from all the authors. I.C. and F.L. led the project.

## Competing interests
The authors declare no competing interests.

**Additional information**

