## [Peer Review File · Communications Chemistry]

Reviewers' comments:

Reviewer #1 (Remarks to the Author):

This article reports (and provides an explanation for) differences in the lifetimes of the excited ($\pi\pi^*$) states accessed by 267 nm photoexcitation of three 'decorated' tryptophan cations: protonated tryptophan (TrpH⁺), sodiated tryptophan (TrpNa⁺) and sodiated alanyl tryptophan (AlaTrpNa⁺). The parent ion of interest is mass selected following preparation by electrospray ionization, excited by a 20 fs duration 267 nm (UV) laser pulse and the resulting photofragment ion mass spectrum measured. In further experiments, the UV pump pulse is followed – at a range of user selected time delays (δt) – by a second 800 nm (IR) probe laser pulse and the fragment ion mass spectrum again measured. Differences in the 'UV pump only' versus 'UV pump plus IR probe' laser induced mass spectra reveal fragment ions arising from IR pumping of population in the $\pi\pi^*$ excited state of the parent ion, and the δt -dependence of this difference signal is used to provide a measure of the lifetime (τ) of the excited state parent ions. τ is shown to be highly species dependent, increasing ~30-fold on changing from TrpH⁺ to TrpNa⁺, and by at least another factor of two in the case of AlaTrpNa⁺. Supporting electronic structure (DFT) calculations provide a persuasive explanation for these observations; the 'chromophore' in all cases is considered to be the aromatic (indole) system, and the role of the appendages is to 'tune' the energy of the (diabatically bound), 'bright' $\pi\pi^*$ state relative and that of a (diabatically unbound), 'dark' $\pi\sigma^*$ excited state that promotes the non-radiative decay (predissociation) of the $\pi\pi^*$ state. That substituents on the central chromophore will 'tune' the coupling between $\pi\pi^*$ and $\pi\sigma^*$ states in heteroaromatic molecules is not new (see e.g. Pino et al., JCP 133, 124313 (2010); Karsili et al., Chem. Sci. 4, 2434 (2013); Wenge et al., PCCP 17,16246 (2015)), but the application to cations is novel, as is the quantification of the ultrafast dissociation timescales. Similarly, the conclusion that the 'bright' $\pi\pi^*$ state of TrpH⁺ has a sub-picosecond lifetime is not new – see e.g. Boyarkin et al., JACS 128, 2816 (2006) and that it can be increased by complexation, see e.g. Mercier et al., JACS 128, 16938 (2006).

Nonetheless, I am content that the manuscript will be of interest to others in the field and is likely to influence thinking sufficiently to merit publication in Communications Chemistry. Prior to publication, however, I would encourage the authors to consider not just the relevance (or not) of the above works but also the following points and act on them as they see fit.

1. Abstract: Line 12 highlights 'a striking increase of the timescale' (of what?). Line 15 reports that 'A simple model is proposed to rationalize the results obtained.' Just a few extra words here outlining the basis of the model might be helpful.

2. Lines 44/45 includes the very imprecise phrase 'and the $\pi\sigma^*$ state is a σ -type orbital located on a specific bond', which will be hard for anyone outside the immediate field to appreciate.

3. Line 92 might benefit from a reference where a reader can confirm the assertion that the m/z 130 feature is not present in statistical dissociation experiments. There seems to be a (small) peak at this m/z in the UV-laser-off spectrum shown in fig. S1.

4. I am more familiar with the underlying photophysics here than most readers will be, but I still had to work quite hard to understand what was going on. Only by referring to the supplementary information (SI) was it clear that fig. 2(a) is the 'difference' between UV 'pump-on' and 'pump-off' spectra. The key feature is the UV photoinduced peak at m/z 130. The UV pulse has a broad energy bandwidth and the parent ions will be formed in a similarly broad spread of internal energy states, so how should one think about this fragment ion formation process? Is it available to all photoexcited molecules, or just that fraction with internal energy concentrated in particular nuclear degrees of freedom? This fragment ion yield is further boosted when the IR pulse (which also has a broad bandwidth) follows the UV pulse at short δt , and it is such IR induced increases in the chosen

fragment ion yields that are shown in figs. 2(b)-2(d). I'm still not sure that I understand what the 'intensity(%)' scale means. Readers might also be helped to have it spelled out that the ion fragmentation process that gives rise to the monitored (e.g. m/z 130) peak simply provides a convenient probe of population excited to the parent $\pi\pi^*$ state; it is not the product of the $\pi\sigma^*$ state mediated dissociation process that is here invoked as determining the $\pi\pi^*$ state lifetime. Whilst making this clearer, it might also help readers understand why the fitting allows A_{step} to be positive or negative – depending on the system studied, or even on which m/z peak one chooses to monitor (in the case of AlaTrpNa+).

5. Line 142 and accompanying text might benefit from making clear that the reported energies are vertical excitation energies from the ground state minimum energy geometry (which is spelled out in the SI).

6. Line 163. I found the text 'For a well-defined molecular system (neutral tryptophan in this study)' confusing. Aren't the systems studied here selected molecular ions?

Reviewer #2 (Remarks to the Author):

In this manuscript, the authors examined the substitution effect on the nonradiative dynamics of the tryptophan derivatives cation which could be a model of peptides. They applied femtosecond UV-IR pump-probe spectroscopy for the mass selected tryptophan derivatives cation, TrpH+, TrpNa+, and AlaTrpNa+, and measured the decay rate of the $\pi\pi^*$ state, where their aim is to reverse the order of the energies of the $\pi\pi^*$ and $\pi\sigma^*$ states, that is if $\pi\pi^*$ is higher than $\pi\sigma^*$ we expect a fast $\pi\pi^* \rightarrow \pi\sigma^*$ IC, while when the order is reversed the IC will not occur. They observed that the $\pi\pi^*$ lifetime of 390 fs for TrpH+ is elongated to 13 ps for TrpNa+ and 35 ps for AlaTrpNa+ as expected from the electronic state energy calculation in Fig. 3. They further calculated the PE curves as a function of N-H and N-Na distance for more theoretical supports and described that ϵ_g is an important factor for control $\pi\pi^* \rightarrow \pi\sigma^*$ IC rate. The control of nonradiative process by the substitution on $\pi\pi^* \rightarrow \pi\sigma^*$ IC and $\pi\pi^* \rightarrow n\pi^*$ has been the subject of experimental and theoretical interests for the aromatic system, and the present results presented an additional aspect that this idea can be also applied to the tryptophan derivatives cation system, and this work is worth to be published.

I have following comments which the authors should consider,

1. The control of nonradiative decay by substitution is extensively performed for cinnamic acids and cinnamate, where the same idea with this work is reported. They should refer such the work, for example, Kinoshita et al, Phys.Chem.Chem.Phys.,2021, 23, 834. They gave more precise analysis by using Marcus theory.
2. It is true that the decay lifetimes of TrpNa+ and AlaTrpNa+, became much longer than TrpH+. Even if it is so, their lifetime is still very fast 13 ps and 35 ps. The authors should comment what kind of other nonradiative process still exists in these cations.
3. It is not clear that the energy levels of Fig.3 are vertical transition energies or adiabatic. I think they should discuss based on the adiabatic energy. They authors should reply.
4. Which transition(s) does the second laser monitor for measuring the decay of the $\pi\pi^*$ state. I suggest an acceptance of the manuscript after the authors considering and replying to above comments.

Reviewer #3 (Remarks to the Author):

The paper by Hervé et. al. reports a study of the ultrafast dynamics of three different systems based on tryptophan chromophore, using a state-of-the-art experimental technique with real time resolution and routine electronic structure calculations (EE) at the TD-DFT level of theory. Moreover, a simplified semi-empirical model, based on electronic structure calculations, is presented in order to describe/classify qualitatively the lifetimes found in certain m/z channels after photon excitation. Regarding the manuscript, I do not find it completely adequately presented. For instance, the lifetime of a fourth system (GlyTrpNa+) shown in the ESI, and is used to support the conclusions, but it is neither mentioned in the manuscript nor included in the computations to see how it fits with the model proposed. Furthermore, in my opinion, the cited bibliography does not provide a good background of the field. Even though the authors present novel experimental results, and the model proposed seems to be interesting due to its simplicity, I believe that the main conclusions do not give a new insight to help move forward the field. The predictions the authors get with their model can be easily and rapidly obtained with standard EE calculations. Besides, I have some serious reservations about the computational method utilized to describe the dynamics of these systems, and the interpretation of the deactivation pathways proposed for them, based on the computational results and the reported results for TrpH+. For all of this, I do not think that this manuscript is suitable for publication in Commun. Chem.

Some considerations:

Regarding the methodology:

The authors use TD-DFT to interpret the dynamics involving electronic excited states of CT character despite the fact that it is well known that this methodology fails in the description of such states, and it is hence not recommended. What is more, there is a considerable amount of bibliography explaining why TD-DFT is not suitable for the description of TrpH+. For instance, see Eur. Phys. J. D 51, 109–116 (2009), Chem. Rev. 2020, 120, 3296–3327 (and references therein), or J. AM. CHEM. SOC. 9 VOL. 129, NO. 19, 2007 (this one used by the authors to compare with their own results). The authors get a $\pi\sigma^*$ state as S1 in agreement with results obtained in a paper which is previous to those mentioned before (J. Am. Chem. Soc. 2006, 128 (51), 16938-16943). Several later publications urge not to use TD-DFT due to the bad description of the energetics of $\pi\sigma^*$ and πCO^* states. This is a very important issue because the authors use the energy gap difference between the $\pi\pi^*$ and $\pi\sigma^*$ states of a pair of systems as a “measurement” of the relative lifetimes of those systems. For the TrpH+ system, the gap the authors obtain is completely wrong and this could be happening to the other systems as well. It is therefore important to see if the same conclusions are still obtained when a method of higher hierarchy and suited for this type of systems, as CC2, is used instead. There are some other points to stress about the energy gap:

i- I believe it is ill-defined in Fig 1b, since the $\pi\sigma^*$ PES is considered as a bound PES and not dissociative as it is commonly found.

ii- It is not a good practice to associate the energy gap obtained at the FC geometry as a measurement of the barrier for the crossing. Besides, some local minimum could be found acting as traps. And it is not a good practice either to use only this criterium to discuss the dynamics of the systems in the electronically excited states. For instance, geometry optimization in the different excited states should be performed to study the character of each state and see in which cases

barriers are developed.

The coupling between the $\pi\pi^*$ and $\pi\sigma^*$ states for TrpH⁺ is obtained between +0.2 and +0.3 eV above the band origin, in qualitative agreement with experimental results. There is a barrierless channel evolving on the $\pi\pi^*$ surface which is observed in the mass channel $m/z=132$ (not 130) and it is not competitive with the other channels which show dynamics in the fs time regime. Moreover, contrary to what the authors say in the manuscript, the energetic of the excited states is highly dependent on the molecular structure of the conformers (see for instance the difference of ~ 0.1 eV between the two most stable conformers of TrpH⁺ reported in J. AM. CHEM. SOC. 9 VOL. 129, NO. 19, 2007).

Regarding the interpretation of the excited state dynamics in TrpH⁺

The authors say the dynamics of the three systems evolve through the $\pi\pi^*$ and $\pi\sigma^*$ PESs. Particularly, they say the fastest de-excitation pathway in TrpH⁺ (~ 400 fs) is attributed to the $\pi\pi^*$ - $\pi\sigma^*$ coupling, whilst the slowest pathway (15 ps) corresponds to the internal conversion after excited-state proton transfer (ESPT). They cite some old papers where the dynamics of TrpH⁺ was studied to support this idea. Nevertheless, there is new bibliography where the deactivation pathways of TrpH⁺ are reinterpreted, the fastest being ascribed to a barrierless H transfer to the C=O group followed by the C α -C β bond cleavage, and the slowest to the H loss via the $\pi\sigma^*$ state due to the barrier developed in the coupling with the $\pi\pi^*$ state, observed experimentally (Eur. Phys. J. D 51, 109–116 (2009), Chem. Rev. 2020, 120, 3296–3327 and references therein). The authors rule out the dynamics via πCO^* state because, according to their calculations, it is very high in energy. Nevertheless, it should be considered as it is one of the main deactivation pathways for TrpH⁺. It is important to recall that the computations systematically overestimate the transition energies. The S₁ state is computed at 4.90 eV and the origin of the band is reported at 4.35 eV, and the energy of a laser pulse used is 4.66 eV. Therefore, the description of the deactivation pathway by the authors is in the best case, incomplete, and it could be different for some of the other systems and change the interpretation of the obtained results.

General considerations.

Finally, I find ambitious the title the authors have chosen for the article because they have used only one chromophore and only two ions. So, I do not recommend using the peptide terminology to show generality unless amino acids of different nature used as chromophores drive to the same conclusions. For instance, I wonder if the model could be applied using Glycine. Or using a different cation as Au⁺, which it has shown to produce a low energy CT state when it is complexed with Trp (Angew. Chem. Int. Ed. 2009, 48, 7829–7832).

Reviewers' comments:

Reviewer #1 (Remarks to the Author):

We wish to thank the reviewer who pointed out a number of aspects that indeed should be clarified in order to make our manuscript accessible for a broad audience. We hope that we have been able to improve our manuscript in that direction. We have added the following references: Pino et al., JCP 133, 124313 (2010); Karsili et al., Chem. Sci. 4, 2434 (2013); Wenge et al., PCCP 17,16246 (2015), Boyarkin et al., JACS 128, 2816 (2006), Mercier et al., JACS 128, 16938 (2006).) which certainly provide a better perspective to our work.

1. Abstract: Line 12 highlights 'a striking increase of the timescale' (of what?). Line 15 reports that 'A simple model is proposed to rationalize the results obtained.' Just a few extra words here outlining the basis of the model might be helpful.

We changed to "A striking increase of the timescale of the $\pi\pi^*$ - $\pi\sigma^*$ dynamics" and "A model is proposed to rationalize the results obtained (GLOSS : global and local stabilization separation). It is based on the localized and delocalized effects of the adduct on the electronic structure of the molecule."

2. Lines 44/45 includes the very imprecise phrase 'and the $\pi\sigma^$ state is a σ -type orbital located on a specific bond', which will be hard for anyone outside the immediate field to appreciate.*

We changed to :

"and the $\pi\sigma^*$ state has a σ character located on a specific X-H bond, with X being usually an heteroatom (N, O or S)"

3. Line 92 might benefit from a reference where a reader can confirm the assertion that the m/z 130 feature is not present in statistical dissociation experiments. There seems to be a (small) peak at this m/z in the UV-laser-off spectrum shown in fig. S1.

We meant that this minor contribution does not affect the conclusion of the paper. Indeed, m/z 130 is not absent from statistical dissociation but represents a minor fragment. This has been corrected in the manuscript, and we have added the references El Aribi et al JPCA 2004 and Lioe et al JASMS 2004 that discussed the small yield of m/z 130 in statistical dissociation.

"m/z=130, which is barely present in statistical dissociation experiments such as Collision-Induced Dissociation (CID)"

4. I am more familiar with the underlying photophysics here than most readers will be, but I still had to work quite hard to understand what was going on. Only by referring to the supplementary information (SI) was it clear that fig. 2(a) is the 'difference' between UV 'pump-on' and 'pump-off' spectra. The key feature is the UV photoinduced peak at m/z 130. The UV pulse has a broad energy bandwidth and the parent ions will be formed in a similarly broad spread of internal energy states, so how should one think about this fragment ion

formation process? Is it available to all photoexcited molecules, or just that fraction with internal energy concentrated in particular nuclear degrees of freedom? This fragment ion yield is further boosted when the IR pulse (which also has a broad bandwidth) follows the UV pulse at short δt , and it is such IR induced increases in the chosen fragment ion yields that are shown in figs. 2(b)-2(d).

I'm still not sure that I understand what the 'intensity(%)' scale means. Readers might also be helped to have it spelled out that the ion fragmentation process that gives rise to the monitored (e.g. m/z 130) peak simply provides a convenient probe of population excited to the parent $\pi\pi^*$ state; it is not the product of the $\pi\sigma^*$ state mediated dissociation process that is here invoked as determining the $\pi\pi^*$ state lifetime. Whilst making this clearer, it might also help readers understand why the fitting allows A_{step} to be positive or negative – depending on the system studied, or even on which m/z peak one chooses to monitor (in the case of AlaTrpNa+).

We agree with the reviewer that the pump-probe scheme and the description of the measured signal needed to be improved in our manuscript, and that some information in the SI should be included in the main text for sake of clarity. We have made the following changes:

In fig 2 caption “a) intensity on the y-axis represents the signal normalized on the parent intensity (i.e. $m/z=227$ is 100%).”

« In the experiment, the pump pulse excites *the parent $\pi\pi^*$ state*, and the probe pulse promotes the population of the excited state to higher lying states, eventually leading to fragmentation. The delay-dependent signal of the m/z 130 photofragment is thus a signature of the dynamics in the $\pi\pi^*$ state. Let us notice that this fragment is simply a convenient signature of the dynamics occurring in the parent $\pi\pi^*$ excited state, as such that the extracted timescale is related to the excited state dynamics and not to the fragmentation process itself.”

“the dynamics is monitored by measuring the delay dependent variation of the fragment signal. To do so the signal obtained following the interaction of the pump (UV) only is subtracted from the signal obtained with the pump and the probe, for each delay.”

5. Line 142 and accompanying text might benefit from making clear that the reported energies are vertical excitation energies from the ground state minimum energy geometry (which is spelled out in the SI).

We have added “ the reported energies are vertical energies from the ground state minimum energy geometry. Although the adiabatic energies usually serve as a reference when considering non-adiabatic dynamics, the choice of vertical energies allows to have the same reference for all molecules even in the case of dissociative states or more complex PES. ”

6. Line 163. I found the text ‘For a well-defined molecular system (neutral tryptophan in this study)’ confusing. Aren’t the systems studied here selected molecular ions?

Indeed this is confusing.

It has been replaced by “for a well-defined molecular system (tryptophan-based ions in this study)”.

Reviewer #2 (Remarks to the Author):

We thank the reviewer for the comments and for pointing out a very interesting recent theoretical work where the effect of chemical substitution on the energetic of pi^* states and on isomerization has been studied. This theoretical work illustrates another control scheme using chemical substitution instead of adding an adduct atom. This other scheme would be very interesting to investigate experimentally. This recent publication shows that these kind of questions are timely and should be further investigated. We have followed the recommendations and improved the discussion regarding other processes involved. We hope the new version will satisfy the reviewer.

I have following comments which the authors should consider,

1. The control of nonradiative decay by substitution is extensively performed for cinnamic acids and cinnamate, where the same idea with this work is reported. They should refer such the work, for example, Kinoshita et al, Phys.Chem.Chem.Phys.,2021, 23, 834. They gave more precise analysis by using Marcus theory.

We have added the reference « Kinoshita et al, Phys.Chem.Chem.Phys.,2021, 23, 834 »

2. It is true that the decay lifetimes of TrpNa⁺ and AlaTrpNa⁺, became much longer than TrpH⁺. Even if it is so, their lifetime is still very fast 13 ps and 35 ps. The authors should comment what kind of other nonradiative process still exists in these cations.

We agree with the reviewer. Although our article is focused on one specific aspect of the photoinduced processes in peptides, other processes might exist in general. We have discussed these aspects in the discussion as follows:

“Although this work focuses on the influence of adduct atoms on $\pi\pi^*$ - $\pi\sigma^*$ dynamics in peptides, the UV induced dynamics in these species is rather complex and many other aspects might be investigated in the future. More generally, other processes such as structural changes, including isomerization, or spin effects (role of triplet states) might also be considered in the perspective of a controlled micro-environment, calling for new experimental investigations.

3. It is not clear that the energy levels of Fig.3 are vertical transition energies or adiabatic. I think they should discuss based on the adiabatic energy. They authors should reply.

We agree with the reviewer that, in order to apply Marcus theory for instance, the adiabatic energies should be used. Here, we used the vertical ones in order to use a common reference of all molecules including those with dissociative states (with no minimum). In any case we only consider the variation of the gap from one molecule to another, not the gap itself. Using the adiabatic energy when it is possible, leads to the same results since it is a signature of the global energy shift of the electronic states.

4. Which transition(s) does the second laser monitor for measuring the decay of the $\pi\pi^*$ state.

The probing process correspond to the absorption of IR photon by the $\pi\pi^*$. This transition leads to higher lying electronic states that subsequently fragment. This means that the observation of the dynamics relies on the efficiency of IR induced fragmentation. This is now explained in more details in the manuscript.

“In the experiment, the pump pulse excites *the parent $\pi\pi^*$ state*, the probe pulse promotes the population of the excited state to higher lying states that eventually lead to fragmentation. The delay dependent signal of the peak at m/z 130 is a signature of the dynamics in $\pi\pi^*$ state. Let us notice that this peak is simply a convenient signature of the dynamics occurring in the parent $\pi\pi^*$ excited state, as such that the extracted timescale is related to the excited state dynamics and not to the fragment”

Reviewer #3 (Remarks to the Author):

We thank the reviewer for his/her comments. We agree that our manuscript can be improved in order to clarify the goal of this study and that some aspects that were discussed in the SI should be in fact included in the text. We wish to stress that the subject of the article is not the detailed understanding of the UV induced dynamics in TrpH^+ . This has already been investigated in details by other colleagues and seminal results have already demonstrated the current limitation in the understanding of this “pathological” case. For a complete overview of the latest results, see review Chem. Rev. 2020, 120, 3296–3327 that contains all the conclusions that are used in our article on TrpH^+ , as mentioned by the referee. What we do here, is to study the evolution of the dynamics from one system to another. The detail of the dynamics for each individual system, above the non-adiabatic couplings occurring on the ultrafast timescale, is beyond the scope of this article. Here we show that a simple model can be used to understand this evolution. The strength of the model is that it allows to interpret the evolution in terms of global and local effect of the adduct atom. This evolution is also reproduced by TDDFT calculations that are not developed to reproduce the individual dynamics but the trend from one molecule to another. Here again, the TDDFT calculation validates our model, which in turn gives physical arguments to the observed trend.

In the following we have answered the reviewer’s comments, point by point. In particular, we performed higher-level calculations (at the CC2 level) that led to the same results and confirmed our analysis. We hope that these answers and the new version of the manuscript will satisfy the reviewer.

... For instance, the lifetime of a fourth system (GlyTrpNa+) shown in the ESI, and is used to support the conclusions, but it is neither mentioned in the manuscript nor included in the computations to see how it fits with the model proposed.

The direct comparison could not be performed because the sodium affinity, a necessary parameter of our model, has never been measured experimentally as far as we know.

Therefore we rather use this example as a perspective in the SI to explain what can be expected from our model.

This is now specified in the SI: “This value is again increased compared to TrpNa⁺, and is due to the higher affinity of the dipeptide compared to bare tryptophan, revealing the strength and possibilities brought by the model.”

... The predictions the authors get with their model can be easily and rapidly obtained with standard EE calculations.

EE calculations will not provide any general insight. The strength of our model is to interpret the process as a combination of a local and a global effect of the adduct atoms which provides a general framework to understand the mechanism with simple parameters as the adduct affinities, instead of performing systematic, raw, calculations.

The authors use TD-DFT to interpret the dynamics involving electronic excited states of CT character despite the fact that it is well known that this methodology fails in the description of such states, and it is hence not recommended. What is more, there is a considerable amount of bibliography explaining why TD-DFT is not suitable for the description of TrpH⁺. For instance, see Eur. Phys. J. D 51, 109–116 (2009), Chem. Rev. 2020, 120, 3296–3327 (and references therein), or J. AM. CHEM. SOC. 9 VOL. 129, NO. 19, 2007 (this one used by the authors to compare with their own results). The authors get a $\pi\sigma^$ state as S1 in agreement with results obtained in a paper which is previous to those mentioned before (J. Am. Chem. Soc. 2006, 128 (51), 16938-16943). Several later publications urge not to use TD-DFT due to the bad description of the energetics of $\pi\sigma^*$ and π_{CO}^* states. This is a very important issue because the authors use the energy gap difference between the $\pi\pi^*$ and $\pi\sigma^*$ states of a pair of systems as a “measurement” of the relative lifetimes of those systems. For the TrpH⁺ system, the gap the authors obtain is completely wrong and this could be happening to the other systems as well. It is therefore important to see if the same conclusions are still obtained when a method of higher hierarchy and suited for this type of systems, as CC2, is used instead. There are some other points to stress about the energy gap:*

It is indeed very well-known that TDDFT can be problematic in the case of CT. In order to check the validity of our interpretation with respect to the accuracy of the theoretical method, we conducted the same calculations using high-level CC2 calculations, with an aug-cc-pVDZ basis (as used in the literature, see e.g., Eur. Phys. J. D 51, 109–116 (2009) and J. AM. CHEM. SOC. 9 VOL. 129, NO. 19, 2007). Overall, the CC2 calculations lead to the same conclusion, with small differences in energetical values for the different states and PESs. We thus replaced all the figures in the main text and SI with these new calculations for the three molecules.

Additionally, results obtained with the previously used TDDFT methodology are kept in the SI, for the dipeptide AlaTrpNa⁺ (see Table S2 and Figure S8). They allow for a comparison with high-level CC2 calculations, showing that this methodology gives reasonable results, and could be further used for larger molecular systems (polypeptides etc), where higher-hierarchy calculations are unaffordable yet. Nonetheless, we specifically mentioned in the SI that failures of TDDFT are known for CT states by adding a reference to Dreuw, A.; Head-Gordon, M. J. Am. Chem. Soc. 2004, 126, 4007-4016.

i- I believe it is ill-defined in Fig 1b, since the $\pi\sigma^$ PES is considered as a bound PES and not dissociative as it is commonly found.*

Fig 1b is just a schematic to illustrate in a simple manner our model. The dissociative character is discussed elsewhere in the manuscript. But we agree that it can be confusing so we added in the caption: “in this figure, the $\pi\sigma^*$ state is represented as a bond state for sake of simplicity while it corresponds to a dissociative state”

ii- It is not a good practice to associate the energy gap obtained at the FC geometry as a measurement of the barrier for the crossing. Besides, some local minimum could be found acting as traps. And it is not a good practice either to use only this criterium to discuss the dynamics of the systems in the electronically excited states. For instance, geometry optimization in the different excited states should be performed to study the character of each state and see in which cases barriers are developed.

Of course, the correct representation of the gap would involve the adiabatic energy. But this is not applicable in the case of a dissociative state. Since our goal is to compare the different molecules therefore we chose a common reference that is the vertical energy. Using the adiabatic energies would lead to the same results as it is the signature of the general energy shift of the state.

Additionally, the choice of the energy gap as a single criterion was motivated by the goal of comparing the dynamics in different systems, and appears to be a good indicator (yet not quantitative of course) of the occurring processes.

This is now mentioned in the manuscript:

“Although the adiabatic energies usually serve as a reference when considering non-adiabatic dynamics, the choice of vertical energies allows to have the same reference for all molecules even in the case of dissociative states or more complex PES”

The coupling between the $\pi\pi^$ and $\pi\sigma^*$ states for TrpH⁺ is obtained between +0.2 and +0.3 eV above the band origin, in qualitative agreement with experimental results. There is a barrierless channel evolving on the $\pi\pi^*$ surface which is observed in the mass channel $m/z=132$ (not 130) and it is not competitive with the other channels which show dynamics in the fs time regime. Moreover, contrary to what the authors say in the manuscript, the energetic of the excited states is highly dependent on the molecular structure of the conformers (see for instance the difference of ~ 0.1 eV between the two most stable conformers of TrpH⁺ reported in J. AM. CHEM. SOC. 9 VOL. 129, NO. 19, 2007).*

As the reviewer mention, the energy position of the different states is highly geometry-dependent. This was already mentioned in our previous text : “The order is however known to be strongly dependent on the chosen computation method and molecular geometry” This explains the difficulty to extract reliable and precise information about the dynamics in TrpH⁺. However our article focuses on the comparison between the systems, not the precise dynamics of TrpH⁺, which thus needs calculations/parameters that enable for an efficient comparison of the different systems, sometimes at the expense of quantitative agreement.

Regarding the interpretation of the excited state dynamics in TrpH⁺

The authors say the dynamics of the three systems evolve through the $\pi\pi^$ and $\pi\sigma^*$ PESs. Particularly, they say the fastest de-excitation pathway in TrpH⁺ (~ 400 fs) is attributed to the $\pi\pi^*$ - $\pi\sigma^*$ coupling, whilst the slowest pathway (15 ps) corresponds to the internal conversion after excited-state proton transfer (ESPT). They cite some old papers where the dynamics of TrpH⁺ was studied to support this idea. Nevertheless, there is new bibliography where the deactivation pathways of TrpH⁺ are reinterpreted, the fastest being ascribed to a barrierless H transfer to the C=O group followed by the C α -C β bond cleavage, and the slowest to the H loss via the $\pi\sigma^*$ state due to the barrier developed in the coupling with the $\pi\pi^*$ state, observed experimentally (Eur. Phys. J. D 51, 109–116 (2009), Chem. Rev. 2020, 120, 3296–3327 and references therein). The authors rule out the dynamics via π_{CO}^* state because, according to their calculations, it is very high in energy. Nevertheless, it should be considered as it is one of the main deactivation pathways for TrpH⁺. It is important to recall that the computations systematically overestimate the transition energies. The S1 state is computed at 4.90 eV and the origin of the band is reported at 4.35 eV, and the energy of a laser pulse used is 4.66 eV. Therefore, the description of the deactivation pathway by the authors is in the best case, incomplete, and it could be different for some of the other systems and change the interpretation of the obtained results.*

The pump photon energy is indeed 4.67 eV, i.e., about 0.30 eV above the origin of the $\pi\pi^*$ state in TrpH⁺ (see Mercier *et al*, JACS 2006), while it is calculated at 4.77 eV in our CC2 calculations. As pointed out by the referee, this means that a significant amount of energy is deposited in the $\pi\pi^*$ state, even being higher than the barrier for H-loss via the $\pi\sigma^*$ state (which is of about 0.2 eV, see Soorkia *et al*, Chem. Rev. 2020). The consequence is that this deactivation pathway is very active in our experiment, although it might not be the only one. Our CC2 calculations also confirm that the π_{CO}^* state remains high in energy compared to the $\pi\pi^*/\pi\sigma^*$ states and the pump photon energy, even when shifted down in energy by ca. 0.4 eV to match experimental values. As a consequence, this state is believed to not intervene in the relaxation pathway, at least in the very first instants of relaxation, because of a stronger coupling with the closer $\pi\sigma^*$ state. It may however play a role in the long-term fragmentation (following $\pi\pi^*$ - $\pi\sigma^*$ transfer), as discussed in Eur. Phys. J. D 51, 109–116 (2009).

General considerations.

Finally, I find ambitious the title the authors have chosen for the article because they have used only one chromophore and only two ions. So, I do not recommend using the peptide terminology to show generality unless amino acids of different nature used as chromophores drive to the same conclusions. For instance, I wonder if the model could be applied using Glycine. Or using a different cation as Au⁺, which it has shown to produce a low energy CT state when it is complexed with Trp (Angew. Chem. Int. Ed. 2009, 48, 7829–7832).

We agree with the reviewer that the title can be misleading we have changed it to “Controlled Ultrafast $\pi\pi^*$ - $\pi\sigma^*$ dynamics in tryptophan based Peptides with Tailored Micro-environment”

We also agree with the reviewer that many other possible systems can be investigated, which will rise other questions. Even if this is beyond the scope of our article, it is our hope

that our article will motivate further experimental investigations in that direction. The article mentioned by the reviewer is co-authored by one the authors of the present work and we agree that further investigation could follow this specific case.

Reviewers' comments:

Reviewer #2 (Remarks to the Author):

Comment on the revised manuscript "Controlled Ultrafast $\pi\pi^*$ - $\pi\sigma^*$ Dynamics in Peptides with Tailored Micro environment" by M. Hervé et al.

Although the authors replied to all my comments, their reply to the comment below is not satisfactory,

My comment "It is not clear that the energy levels of Fig.3 are vertical transition energies or adiabatic. I think they should discuss based on the adiabatic energy. They authors should reply."

They replied as

"Using the adiabatic energy when it is possible, leads to the same results since it is a signature of the global energy shift of the electronic states."

Why can the authors say easily that the order of the adiabatic energy and the vertical energy show the same trend. Since the potential curve (shape) of each electronic state sensitively changes by the environment, it is not always the case that potential curve crossing follows the trend of the vertical energies. Since they are discussing the small energy difference of 0.1 ~ 0.3 eV, careful discussion will be necessary. For example, 0.3 eV is less than the energy of C-H and O-H stretching vibration.

The authors should make this point more clear.

Reviewer #3 (Remarks to the Author):

After reading the new version of the manuscript and SI I think the authors have done a very nice improvement of this work. I find the manuscript more complete and clearer now. Nevertheless, I still have a big concern regarding the computational results on which the authors support their model. Though the authors performed single point calculations at a higher level of theory (CC2), there still is a considerable qualitative discrepancy between the ordering of the excited states of TrpH⁺ obtained in this work and the results reported and reaffirmed in previous works (Chem. Rev. 2020, 120,3296–3327 - J. AM. CHEM. SOC. 9 VOL. 129, NO. 19, 2007). The previously reported energy gaps between the $\pi\pi^*$ states and the $\pi\sigma^*$ are 140 and 400 meV for conformers A and B, respectively (computed at the CC2//MP2/aug-cc-PVDZ-SV(P) level of theory). However, the authors compute an energy gap of -80 and -100 meV for conformers A and B, respectively, with CC2/aug-cc-PVDZ//B3LYP/6-311+G**. It could be possible that the difference comes from the fact that the vertical excitations are computed using DFT relaxed geometries instead of MP2 minima or it could be that the relaxed geometries are not true minima.

This discrepancy is something significant not only because the results obtained here for TrpH⁺ are in a qualitative contrast with previously published results, predicting a barrierless dark state though a barrier has been observed both computationally and experimentally, but also in the context of the model presented here. For instance, if we take the previously reported energy gaps for TrpH⁺ to calculate the delta energy gap ($\delta\epsilon_{g}$) with TrpNa⁺, the results obtained are $\delta\epsilon_{g} = 90$ meV and $\delta\epsilon_{g} = -170$ meV, for conformers A and B, respectively. The $\delta\epsilon_{g}$ calculated for the conformer A differs considerably from the results obtained using the GLoSS model but still are in the same trend. Most important is that the $\delta\epsilon_{g}$

calculated for conformer B (which could be also present in the system) is negative and therefore it would lead to a misinterpretation based on the GLOSS model. Now, the fact is that the dynamics of TrPH⁺ is indeed faster than the dynamics of TrPNa⁺. It is interesting to note that a higher ϵ_{g} for TrPH⁺ compared to TrPNa⁺ tells nothing conclusive about the deactivation dynamics of these systems as the barrier for the $\pi\pi^*/\pi\sigma^*$ crossing could still be lower for TrPH⁺ or could even be higher but tunneling could play a significant role in TrPH⁺ making the transfer faster than the case for TrPNa⁺ where no tunnel effect is expected. I understand the idea of this work is to describe general trends. However, I think the authors need to review and clarify this point before sending this article to publish to assure their model does not conduct misinterpretations.

Reviewers' comments:

Reviewer #2 (Remarks to the Author):

Comment on the revised manuscript "Controlled Ultrafast $\pi\pi^*$ - $n\sigma^*$ Dynamics in Peptides with Tailored Micro environment" by M. Hervé et al.

Although the authors replied to all my comments, their reply to the comment below is not satisfactory,

My comment "It is not clear that the energy levels of Fig.3 are vertical transition energies or adiabatic. I think they should discuss based on the adiabatic energy. They authors should reply."

They replied as

"Using the adiabatic energy when it is possible, leads to the same results since it is a signature of the global energy shift of the electronic states."

Why can the authors say easily that the order of the adiabatic energy and the vertical energy show the same trend. Since the potential curve (shape) of each electronic state sensitively changes by the environment, it is not always the case that potential curve crossing follows the trend of the vertical energies. Since they are discussing the small energy difference of 0.1 ~ 0.3 eV, careful discussion will be necessary. For example, 0.3 eV is less than the energy of C-H and O-H stretching vibration.

The authors should make this point more clear.

We agree with the referee that our statement cannot be considered as a general rule. Depending on the shape of the potential energy surface, the trend can very well be different for the vertical and adiabatic energies.

It turns out that in the case discussed here, the trend is the same which allows to use the two references indifferently and that is what we meant.

In the SI we have included the energy gap taken at the crossing in addition to the vertical energy gap showing that they evolve both similarly.

We also added the following comments.

"Additionally, Table S5 gives the position and energy barrier of the crossing point between the $\pi\sigma^*$ and $\pi\pi^*$ states, where the energy barrier is defined with respect to the minimum of the $\pi\pi^*$ PES. This crossing point is estimated using the PES of Figure 4.b of the main text and Figure S7.b. As observed in Table S5, the increase of the vertical energy gap between the two states correlates with an increase of the energy barrier involved in the dynamics, confirming the vertical energy gap as an equivalent criterion in the case of the present dynamics. "

And in tableS5:

"in the two first columns. The crossing energy, with respect to the minimum of the PES of the lowest $\pi\pi^*$ state, is also given together with its N-Na distance, taken as the energy and N-Na bond length at the crossing point between the $\pi\sigma^*$ and $\pi\pi^*$ states on Figure 4.b (main text) and Figure S7.b."

Reviewer #3 (Remarks to the Author):

After reading the new version of the manuscript and SI I think the authors have done a very nice improvement of this work. I find the manuscript more complete and clearer now. Nevertheless, I still have a big concern regarding the computational results on which the authors support their model. Though the authors performed single point calculations at a higher level of theory (CC2), there still is a considerable qualitative discrepancy between the ordering of the excited states of TrpH⁺ obtained in this work and the results reported and reaffirmed in previous works (Chem. Rev. 2020, 120,3296–3327 - J. AM. CHEM. SOC. 9 VOL. 129, NO. 19, 2007). The previously reported energy gaps between the $\pi\pi^*$ states and the $n\sigma^*$ are 140 and 400 meV for conformers A and B, respectively (computed at the CC2//MP2/aug-cc-pVDZ-SV(P) level of theory). However, the authors compute an energy gap of -80 and -100 meV for conformers A and B, respectively, with CC2/aug-cc-pVDZ//B3LYP/6-311+G**. It could be possible that the difference comes from the fact that the vertical excitations are computed using DFT relaxed geometries instead of MP2 minima or it could be that the relaxed geometries are not true minima.

This discrepancy is something significant not only because the results obtained here for TrpH⁺ are in a qualitative contrast with previously published results, predicting a barrierless dark state though a barrier has been observed both computationally and experimentally, but also in the context of the model presented here. For instance, if we take the previously reported energy gaps for TrpH⁺ to calculate the delta energy gap ($\delta\varepsilon_g$) with TrpNa⁺, the results obtained are $\delta\varepsilon_g = 90$ meV and $\delta\varepsilon_g = -170$ meV, for conformers A and B, respectively. The $\delta\varepsilon_g$ calculated for the conformer A differs considerably from the results obtained using the GLoSS model but still are in the same trend. Most important is that the $\delta\varepsilon_g$ calculated for conformer B (which could be also present in the system) is negative and therefore it would lead to a misinterpretation based on the GLoSS

model. Now, the fact is that the dynamics of TrpH⁺ is indeed faster than the dynamics of TrpNa⁺. It is interesting to note that a higher ε_g for TrpH⁺ compared to TrpNa⁺ tells nothing conclusive about the deactivation dynamics of these systems as the barrier for the $\pi\pi^*/n\sigma^*$ crossing could still be lower for TrpH⁺ or could even be higher but tunneling could play a significant role in TrpH⁺ making the transfer faster than the case for TrpNa⁺ where no tunnel effect is expected. I understand the idea of this work is to describe general trends. However, I think the authors need to review and clarify this point before sending this article to publish to assure their model does not conduct misinterpretations.

We thank the referee for his/her work on our manuscript and interesting discussion. In order to clarify the point mentioned by the referee we have compared the results of excited state calculations performed on various TrpH geometries, i.e., either our geometry obtained using DFT/B3LYP and the geometry obtained in the previous approach from Gregoire et al JACS2007 using MP2, and in both geometries with the different basis set mentioned, i.e. RICC2 aug-cc-pVDZ (our calculations) or RICC2 aug-cc-pVDZ-SV(P) (Gregoire et al).

As discussed in the references mentioned, there is a strong effect of the basis.

Using the basis set and geometry of JACS2007 we obtained the same results described by the referee (in contradiction with our results).

However, the RICC2 aug-cc-pVDZ calculations (larger basis, aug-cc-pVDZ on all atoms) show that all the geometries lead to the same ordering of the states that is the one presented in our article (including the geometry discussed in JACS2007) and validate our previous conclusion.

This comes from the fact that this basis is large and is able to take into account for diffuse orbitals for H that are important for the description of π σ^* state (NH bond), while this effect is less important for the description of the π π^* state. Overall, our new calculations, including the one performed with different geometries still lead to the same conclusion and further consolidate our conclusion. These new results are included and discussed in the SI.

We have added the following:

"To investigate the discrepancies in the state ordering, we performed excited state calculations on various TrpH⁺ geometries, i.e., our geometry obtained using DFT methodology and the geometry taken from ref 3, obtained using MP2 calculations. In both geometries, RICC2 calculations of the excited states were performed using either (full) aug-cc-pVDZ basis set, as employed in the present work, or aug-cc-pVDZ(N,O)+def2-SV(P)(C,H), used in ref 3. The results are shown in Table S2. As shown, both geometries lead to similar excited states. Especially, calculations using the smaller aug-cc-pVDZ(N,O)+def2-SV(P)(C,H) basis set give the same energies and states as in ref 3. The difference between the present calculations and the results in ref 3 therefore comes from the larger basis set used in the present work, which is more suitable to describe the excited states involved in the process. Indeed, the present large basis set takes into account a larger number of diffuse orbitals for C, N and H, that are important in the description of the $\pi\sigma^*$ states."

REVIEWERS' COMMENTS:

Reviewer #3 (Remarks to the Author):

I am in favour of the publication of this article.

Just a general comment, it would be interesting to understand how a barrier is developed in the $\pi\pi^*$ to $\pi\sigma^*$ dynamics of TrPH+ to breach the gap between the experimental and computational findings.

REVIEWERS' COMMENTS:

Reviewer #3 (Remarks to the Author):

I am in favour of the publication of this article.

Just a general comment, it would be interesting to understand how a barrier is developed in the $\pi\pi^*$ to $\pi\sigma^*$ dynamics of TrPH+ to breach the gap between the experimental and computational findings

We thank the reviewer for his/her feedback. We agree that further investigations will help to access a precise description of the mechanistic. Although this is beyond the scope of the present article, we have add the following sentence page6 of the SI:

“Further basis set studies would be interesting to understand the precise $\pi\sigma^*$ - $\pi\pi^*$ ordering in comparison with experimental measurements”